# Visualizing inflammation with an M1 macrophage selective probe via GLUT1 as the gating target

Heewon Cho [1,9], Haw-Young Kwon[2,9], Amit Sharma [3], Sun Hyeok Lee[1], Xiao Liu[4], Naoki Miyamoto[2], Jong-Jin Kim [5], Sin-Hyeog Im [3,6,7], Nam-Young Kang[8] & Young-Tae Chang [1,2,4] ✉

Macrophages play crucial roles in protecting our bodies from infection and cancers. As macrophages are multi-functional immune cells, they have diverse plastic subsets, such as M1 and M2, derived from naïve M0 cells. Subset-specific macrophage probes are essential for deciphering and monitoring the various activation of macrophages, but developing such probes has been challenging. Here we report a fluorescent probe, CDr17, which is selective for M1 macrophages over M2 or M0. The selective staining mechanism of CDr17 is explicated as Gating-Oriented Live-cell Distinction (GOLD) through over-expressed GLUT1 in M1 macrophages. Finally, we demonstrate the suitability of CDr17 to track M1 macrophages in vivo in a rheumatoid arthritis animal model.

Macrophages are critical immune cells and control immune responses by secreting cytokines and engaging other immune cells to protect our bodies. The remarkable feature of macrophages is their plasticity, which allows the polarization of the resting state of macrophage (M0) into different activated states responding to the diverse environment[1]. Macrophages are commonly classified into two representative polarized states: M1 (pro-inflammatory) and M2 (anti-inflammatory) macrophages[2]. Considering their plasticity, developing fluorescent probes to detect specific subpopulations of macrophages will be highly desirable. M1 macrophages are characterized by producing pro-inflammatory cytokines and exhibiting microbicidal activity[3]. However, the continuous over-activation of pro-inflammatory functions of M1 macrophages will lead to chronic inflammation and the development of autoimmune diseases, such as rheumatoid arthritis[4].

To visualize macrophages, non-invasive fluorescent probes have been developed with minimal disturbance towards cellular functions[5,6]. Most reported probes indirectly detect macrophages by targeting reactive molecules[7,8], phagocytic activities in acidic pH[9], or enzymatic activities[10]. For the direct method, our group previously developed CDg16, which gives selectivity to activated macrophages (staining both M1 and M2) over the resting state of M0 macrophages through SLC18B1 transporter as the biomarker[11]. Although the reported probes demonstrate usefulness in biological applications, they usually do not present the capacity to distinguish different subsets of macrophages.

To develop an M1 selective probe, we design a fluorescent library to exploit the different metabolisms of M1 and M2. Whereas M1 macrophages mainly use glucose transporters (GLUTs) to facilitate carbohydrate uptake to sustain aerobic glycolysis, M2 macrophages prefer to take fatty acids by CD36 to feed the TCA cycle and oxidative phosphorylation (OXPHOS) (Fig. 1A)[12]. We construct a Luminescent-Carbohydrate (LC) library based on these metabolic differences, which

[1]School of Interdisciplinary Bioscience and Bioengineering, Pohang University of Science and Technology (POSTECH), Pohang 37673, Republic of Korea. [2]Center for Self-assembly and Complexity, Institute for Basic Science (IBS), Pohang 37673, Republic of Korea. [3]Department of Life Sciences, Pohang University of Science and Technology (POSTECH), Pohang 37673, Republic of Korea. [4]Department of Chemistry, Pohang University of Science and Technology (POSTECH), Pohang 37673, Republic of Korea. [5]Department of Biology, Sunchon National University, Sunchon 57922, Republic of Korea. [6]ImmunoBiome Inc., Pohang 37666, Republic of Korea. [7]Institute for Convergence Research and Education in Advanced Technology, Yonsei University, Seoul 03722, Republic of Korea. [8]Department of Convergence IT Engineering, Pohang University of Science and Technology (POSTECH), Pohang 37673, Republic of Korea. [9]These authors contributed equally: Heewon Cho, Hwa-Young Kwon. ✉e-mail: ytchang@postech.ac.kr

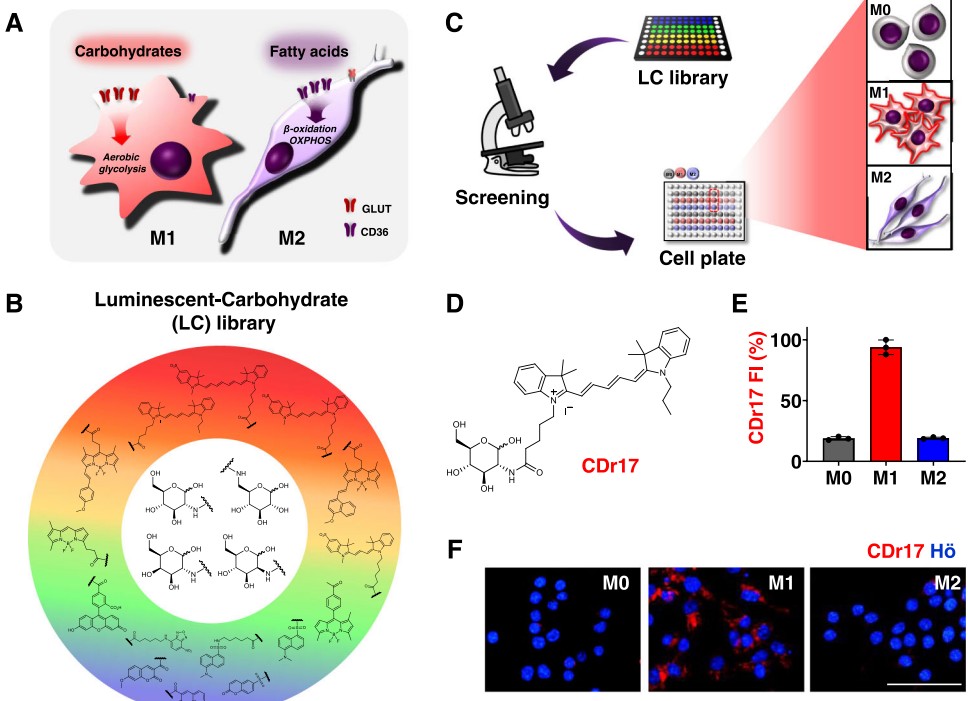

**Fig. 1 | Discovery of M1 selective probe by image-based screening. A** Metabolic pathways of M1 and M2 macrophages. Whereas M1 macrophages upregulate glucose transporters (GLUTs) to uptake carbohydrates and operate aerobic glycolysis, M2 macrophages highly expressed CD36 to make an energy from OXPHOS and β-oxidation. **B** 80-membered Luminescent-carbohydrate (LC) library was composed of 4 backbones of carbohydrates conjugated with 20 different physical properties of fluorophores. **C** Workflow of image-based screening to develop an M1 selective probe. RAW264.7 cells were used as the resting state of macrophages and differentiated into M1 and M2 macrophages. M0, M1, and M2 cells were seeded into 96-well plates, and screened by high-throughput screening microscope. **D** Chemical structure of CDr17. **E, F** Selectivity of CDr17 to M1 over M0 or M2 macrophages from RAW264.7 and relative fluorescent intensity (FI) of CDr17. All the images were acquired at ×40 magnification. Data are presented as mean values ± SD (*n* = 3). All error bars represented standard deviation from three independent measurements. Scale bar, 50 μm. Source data are provided as a Source data file.

aims to target GLUTs (Fig. 1B). Through an unbiased image-based screening of M1 and M2 along with M0 using LC library compounds, we identify an M1-specific probe, CDr17, and demonstrate its practicability to visualize M1 macrophages in the rheumatoid arthritis animal model. Thus, we report the development of an M1-specific fluorescent probe for the visualization of inflammation in vivo.

## Results

### Development of an M1-specific probe

The primary strategy to develop an M1-specific probe is establishing a carbohydrate-based library. In recent decades, researchers have worked with fluorescently labeled carbohydrates as GLUTs substrates[13,14]. Most probes have been utilized in tumor imaging, assuming increased carbohydrate uptake[15–18], while there are very few examples of subtype distinction of immune cells. For an M1 selective probe discovery, we constructed LC library with four carbohydrate backbones conjugated with twenty different fluorophores, hypothesizing that the physical properties of fluorophores are also important to the selectivity for the GLUT target as well as the carbohydrate configuration (Supplementary Table. 1, Supplementary Fig. 1).

For the cell screening system, M1 and M2 macrophages were differentiated from M0, the resting state of RAW264.7, a mouse cell line. The polarized cells were characterized by immunocytochemistry (ICC) and gene expression of cell type-specific biomarkers (Supplementary Fig. 2). M0, M1, and M2 macrophages were seeded into 96 well plates, and treated with LC compounds (Fig. 1C). After 1 h incubation, the cell images were taken by an automated fluorescence microscope. By quantifying the fluorescent intensity of each cell type, a selectivity index (SI) of M1 over M0 or M2 macrophages was calculated, and the result was summarized by a heat map (Supplementary Fig. 3). The

selectivity trend showed a general preference for cyanine dyes to M1 cells in comparison to other fluorophores. Among cyanine dyes, Cy5-derivatives showed a higher selectivity to M1 cells than Cy3-derivatives (Supplementary Fig. 4). Along with fluorophores, carbohydrate backbone structures also influenced the selectivity, presenting that glucose analogs were more preferable to target M1 macrophages than galactose and mannose analogs. The best compound, CDr17 (Compound Designation red 17, Fig. 1D), consists of glucose conjugated with cy52 fluorophore at 2-position (slightly better than in 6-position), showing 5-fold stronger intensity in M1 than M0 or M2 macrophages (Fig. 1E, F). The M1 macrophage selectivity of CDr17 was double confirmed by co-staining with CDg16, which stains both M1 and M2 macrophages without discrimination (Supplementary Fig. 5). The analysis results clearly showed that the M1 cell selectivity is contributed by both carbohydrate configuration and fluorescent motif properties.

### Target identification of CDr17

Motivated by the structure of CDr17, we investigated whether CDr17 selectivity is related to GLUTs, which are regarded as the intracellular carbohydrate uptake machinery. Firstly, a glucose competition experiment was demonstrated in M1 macrophages. The competition assay was performed by D/L-glucose, assuming that CDr17 will be only affected by D-glucose, but not by L-glucose. The cells were incubated with various concentrations of D/L-glucose in glucose-free media for 5 min, followed by CDr17 treatment. The CDr17 intensity decreased by D-glucose in a dose-dependent way, but L-glucose did not influence CDr17 uptake by M1 macrophages (Fig. 2A). In parallel, the inhibition study of GLUTs was performed by a general GLUTs inhibitor, cytochalasin B[19]. After 30 min incubation of M1

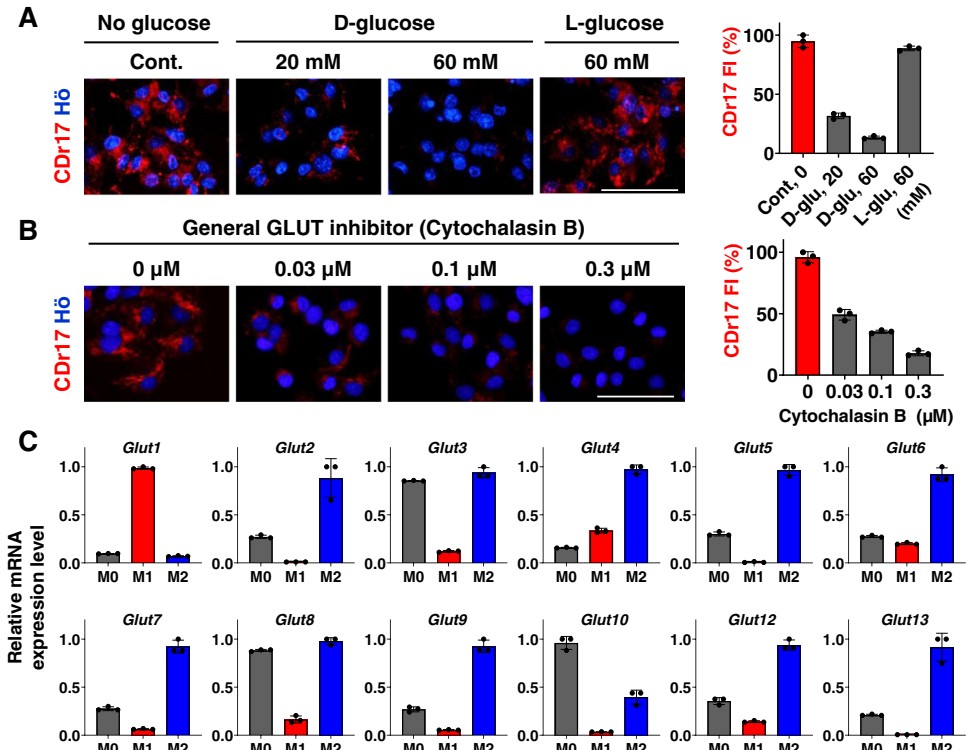

**Fig. 2 | Target identification of CDr17. A** Glucose competition assay was carried out by dose-dependently treated D- and L-glucose with CDr17 (1 μM) in M1 cells from RAW264.7. **B** Inhibition study of GLUTs family by a general GLUT inhibitor (cytochalasin B) with CDr17 (1 μM) in M1 macrophages from RAW264.7. The numeric value of CDr17 fluorescent intensity (FI) was plotted into the graph. All the images were acquired at 40x magnification. Data are presented as mean values ± SD ($n = 3$). All error bars represented standard deviation from three independent measurements. Scale bar, 50 μm. **C** GLUTs gene level from M0, M1 and M2 macrophages from RAW264.7. Data pooled from three individual experiments. Data are analyzed with three samples over three independent experiments. All bar graphs show the mean ± SD ($n = 3$). Source data are provided as a Source data file.

macrophages with varying concentrations of the inhibitor, CDr17 was treated in the cells (Fig. 2B). The outcome showed that cytochalasin B significantly blocked CDr17 uptake through GLUTs in a dose-dependent manner. These results together confirmed that CDr17 mimicked D-glucose, presumably by acting as the substrate of GLUTs.

Next, we compared the performance of CDr17 with 2-NBDG, which is a standard fluorescent probe to monitor glucose uptake through GLUTs[20], to check its selectivity in subsets of macrophages. M0, M1, and M2 macrophages derived from RAW264.7 were stained with 1 μM of 2-NBDG under the same experimental conditions of CDr17 (Supplementary Fig. 6). Since the 2-NBDG signal was too dim, we had to increase the concentration of 2-NBDG up to 100 μM, where 2-NBDG could barely reach the CDr17 performance, requiring almost 100-fold higher concentration than CDr17. For further examination of the probe universality, CDr17 and 2-NBDG were applied to the human-origin macrophage cell line, THP-1. The characteristics of differentiated macrophages were confirmed by antibodies and gene expressions of specific biomarkers (Supplementary Fig. 7). We found that 0.5 μM of CDr17 was good enough to distinguish M1 over M0 and M2 macrophages (Supplementary Fig. 8). In contrast to CDr17, 2-NBDG could not clearly distinguish M1 and M2 macrophage cells at any concentration (Supplementary Fig. 9). The results clearly demonstrated the superior work and universality of CDr17 as the M1 selective probe, reflecting the unique contribution of the Cy5 fluorophore.

To narrow down the CDr17 target among GLUTs, we analyzed the gene expression levels of all known 13 mouse GLUTs one by one in M0, M1, and M2 macrophages (Fig. 2C), along with sodium-glucose transporters (SGLTs) which are also relevant to carbohydrates uptake (Supplementary Fig. 10). Interestingly, the results showed that only GLUT1 was highly expressed in M1 compared to M0 and M2 macrophages. This selective expression of GLUT1 in M1 was confirmed by

GLUT1 antibody through immunocytochemistry (Supplementary Fig. 11). Then, we studied the relationship between the M1 polarization degree and CDr17 or GLUT1 expression (Supplementary Fig. 12). We followed the intensity of CDr17, anti-GLUT1 and anti-CD86 from the same areas at each polarizing state to M1 macrophages. Both CDr17 and GLUT1 showed a strong correlation in the progress of M1 differentiation. With circumstantial evidence, we hypothesized that CDr17 hijacks GLUT1 to enter M1 macrophages.

**Target validation of CDr17**

To confirm whether GLUT1 is indeed the gating target of CDr17, we first tried to block the function of GLUT1 by a GLUT1 selective inhibitor, STF-31[21] (Fig. 3A), and the inhibitor effect on CDr17 uptake was monitored. The entry of CDr17 to M1 cells was remarkably reduced by STF-31 treatment in a dose-dependent manner (Fig. 3B). For further validation, GLUT1 KO (knockout) was carried out via CRISPR/Cas9 in M1 macrophages (Fig. 3C, D). The KO M1 cells showed an 80% decrease in CDr17 intensity compared to control M1 macrophages (Fig. 3E). To further ensure GLUT1 as the molecular target of CDr17, we also generated activated (CRISPRa) and inhibited (CRISPRi) GLUT1 expression in HeLa cell by using CRISPR/dCas9-VPR[22] and -KRAB[23], respectively (Supplementary Fig. 13). As expected, the CRISPRa-GLUT1 cells were brighter, and the CRISPRi-GLUT1 cells were dimmer than control Hela cells by CDr17 treatment. Conclusively, the selective staining mechanism of CDr17 is Gating-Oriented Live-cell Distinction (GOLD), through membrane transporter, GLUT1 as the gating target.

**Visualization of inflammation by CDr17**

Inflammation is a well-known disease symptom affected by M1 macrophages. Before the animal experiment, first we confirmed if CDr17 could be utilized in murine primary cells. Peritoneal macrophages

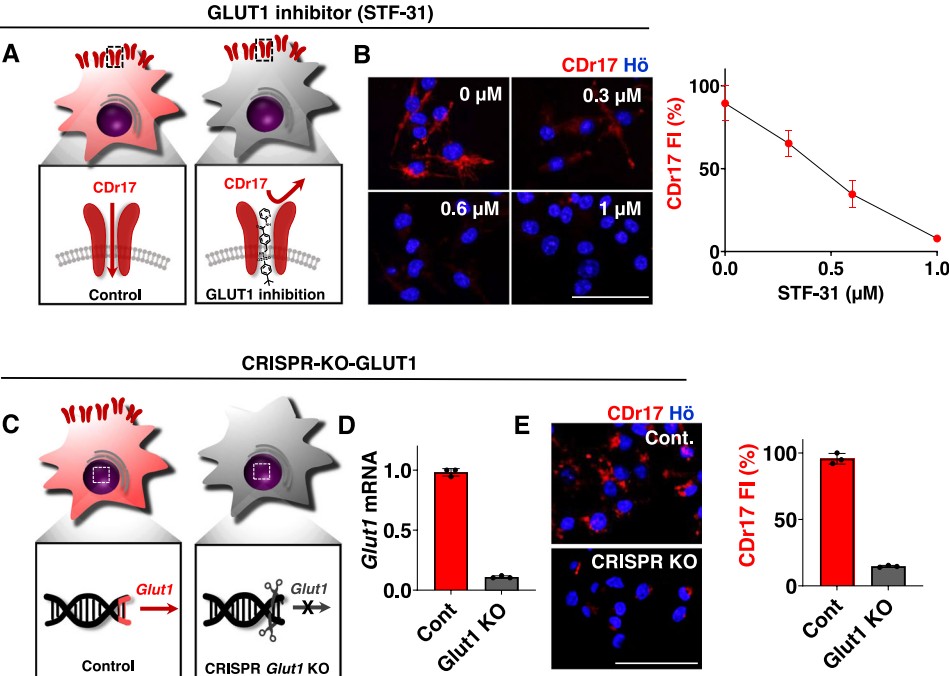

**Fig. 3 | Target validation of CDr17. A** The scheme of GLUT1 inhibition study with GLUT1 specific inhibitor, STF-31. **B** The effects of STF-31 on the uptake of CDr17 by images using M1 macrophages from RAW264.7. STF-1 was added to M1 macrophages, and CDr17 (1 μM) was treated for 30 min. CDr17 fluorescent intensity (FI) was plotted into the graph. All the images were acquired at 40x magnification. Data are presented as mean values ± SD. All error bars represented standard deviation from three independent measurements. **C** Schematic view of GLUT1-CRISPR-knockout (KO) in M1 macrophages. **D** GLUT1 gene level from control and GLUT1 KO M1 cells. Data pooled from three individual experiments. Data are analyzed with three samples over three independent experiments (n = 3). **E** The fluorescence intensity of control and GLUT1 KO M1 macrophages. CDr17 (1 μM) was added for 30 min. All the images were acquired at ×40 magnification. Data are presented as mean values ± SD (n = 3). All the error bars represented standard deviation from three independent measurements. All the scale bars represented 50 μm. Source data are provided as a Source data file.

were harvested from the peritoneal cavity of the mouse, and M0, M1, and M2 macrophages were prepared[24,25]. CDr17 clearly allowed us to distinguish M1 from M0 and M2 macrophages even in the primary cells (Supplementary Fig. 14). Next, we tested whether CDr17 systemically detects the inflamed region in the whole body with acute inflammation models. LPS (5 μg) was injected into the right hind paw for inflammation induction and the left hind paw was injected by PBS as the control. After 24 h, the redness and swelling were observed in the LPS-injected area, but not in the PBS region. After inflammation was formed, CDr17 was applied through intravenous (i.v.) injection. After 30 min, we collected tissues from PBS- and LPS-injected regions. The number of CDr17-stained cells was much higher in LPS-injected tissues than in PBS control region. The stained cells were M1 macrophages, validated by anti-CD86 signal (Supplementary Fig. 15).

Encouraged by the performance of CDr17 in vivo imaging, we expanded its practicality test to rheumatoid arthritis (RA) model. RA is an inflammatory autoimmune disease defined by joint erosion and synovial inflammation[26]. It is known that the majority of macrophages in RA consist of M1 and few M2 macrophages[27]. In light of this, we investigated whether CDr17 could be applied for RA diagnosis. The RA model from mice was established by collagen antibody cocktail method[28] (Fig. 4A). After 3 days of antibody injection (i.v.), LPS was treated to mice by intraperitoneal (i.p.) injection, and then we observed joint swelling and redness after 10 days (Fig. 4B, red arrow). We injected CDr17 (1 mM, 100 μL) into the established model, and collected in vivo images at different time points. The intensity showed the highest peak at 15 min, proving that CDr17 significantly distinguished inflamed region from control in a short time (Fig. 4C). To double check the CDr17 selectivity in mimicking more natural RA, collagen-induced arthritis (CIA) models were prepared. (Supplementary Fig. 16). Once the joint swelling and redness (red arrow) were observed, CDr17 (1 mM, 100 μL) was intravenously injected.

Similar to the CAIA results, CDr17 was able to distinguish RA from the normal. Next, we analyzed the relationship between CDr17 intensity and swelling joint grade. The result clearly demonstrated the positive correlation between swelling grade and CDr17 intensity (Fig. 4D) Then, we monitored CDr17 signal, injecting the probe repeatedly to the same animals (Supplementary Fig. 17). On 12 day, we started to inject CDr17 over time after the illness was induced entirely, and tracked its intensity until 28 day. We observed the decline of swelling grade as days went by, and it was reflected by CDr17 signal decrease. It also proved that CDr17 did not accumulate to the site, despite the repeated injections to the same animals. These results casted the possibility of CDr17 as a diagnostic probe tracking the disease severity. We collected the tissues from the control and RA-induced mouse to confirm whether the signal comes directly from macrophages. (Fig. 4E, F). The images displayed that CDr17 can distinguish RA from the normal, and demonstrated that the stronger intensity of CDr17 came from macrophages which showed higher GLUT1 expression in the synovium tissues compared with B cells and dendritic cells (Supplementary Fig. 18). To make the point clearer, we isolated B cells, dendritic cells, and T cells from the mouse, and analyzed their GLUT1 expression levels including M0 and M1 macrophages derived from RAW264.7. (Supplementary Fig. 19). It also supported that M1 macrophages express higher GLUT1 than other cell types. Collectively, we demonstrated that CDr17 is capable of visualizing inflamed joint region in RA in vivo, along with the possibility of CDr17 as a diagnostic probe.

## Discussion

With the important role of M1 macrophages in autoimmune diseases[29], the need for selective imaging of M1 is emerging. To address this challenging problem, we designed a carbohydrate-based LC library to develop an M1 selective probe, utilizing the high uptake of

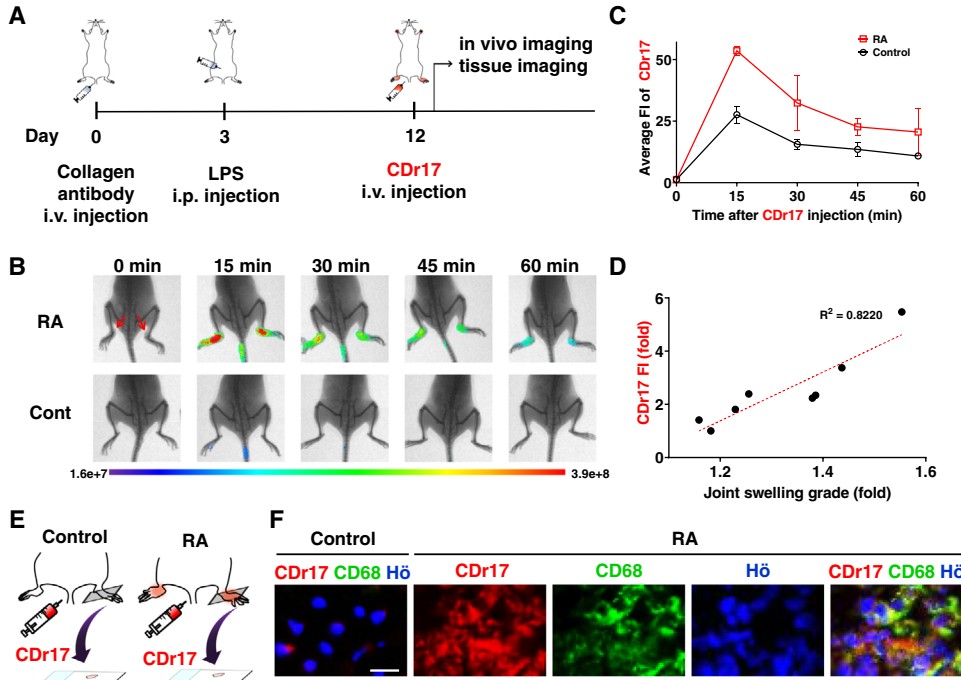

**Fig. 4 | Visualizing rheumatoid arthritis mouse with CDr17. A** Workflow of rheumatoid arthritis (RA)-induced mice preparation. **B** In vivo fluorescence images of RA and control at different time post-injection of CDr17 and **C** the average fluorescent intensity (FI) of CDr17 was plotted into the graph. Data are presented as mean ± SD. ($n = 3$, biologically independent mice per group). **D** Correlation between joint swelling grade and fluorescent intensity of CDr17 after 15 min injection in RA. **E** Scheme of tissue imaging post i.v. injection of CDr17. **F** CDr17 strongly stained macrophages in RA-induced mice, confirmed by anti-CD68. Scale bar, 20 μm. All the images were acquired at ×20 magnification. Data are presented as mean values ± SD ($n = 3$). Error bars: standard deviation from three independent measurements. Scale bar, 50 μm. Source data are provided as a Source data file.

carbohydrates in M1 macrophages. While carbohydrate structure was assumed to be essential in this context, interestingly we found the fluorophore structure was more critical to the M1 selectivity. As the combination result, CDr17, which is composed of Glucose scaffold and 2-position Cy5 fluorophore, has the best selectivity to M1 over M0 and M2 macrophages.

Among possible carbohydrate transporters, we identified the molecular target of CDr17 as GLUT1. GLUT1 was validated to be highly expressed in M1 than M0 and M2 macrophages, and CDr17 utilizes GLUT1 to enter M1 macrophages. Our group classified this transporter-dependent mechanism for cell-specific probe as Gating Oriented-Live cell Distinction (GOLD)[30]. Conventionally, the small molecular probe is assumed to have a selective binding target such as a protein bio-marker, following Protein Oriented-Live cell Distinction (POLD) mechanism[31]. This is a similar concept of the magic bullet of ideal drug molecules suggested by Paul Ehrlich[32]. However, we have observed the situations that the probes may not have a proper binding target of biological macromolecules, but still exist at high concentration in the target cells. GOLD would be a surrogate mechanism to explain and achieve such a cell selectivity using transporters as the catalytic bio-markers. These results suggested that the identification of distinctively expressed gating target could be an excellent strategy for live cell discrimination and GOLD probe design.

The initiation and maintenance of RA are affected by the activity of macrophages, especially M1 macrophages[33]. The quantification of M1 macrophages is directly correlated with the severity of the inflammation[27], and CDr17 can provide the relevant key information by simple fluorescence intensity. The current RA treatment needs significant improvement with 40% unresponsiveness at the moment[34]. Accordingly, an efficient new treatment is needed, and M1 macrophages are one of the most promising targets for this. In our view, CDr17 could play a critical role as an efficient screening platform during the development of therapeutics. CDr17 could potentially assess drug efficacy by monitoring the change of the M1 population upon treatment. As the roles of M1 macrophages become more and more important, we believe that CDr17 will significantly contribute to advancing the research in this field.

## Methods

### Ethical statement

Our research complies with all relevant ethical regulations. Animals were provided by the Pohang University of Science and Technology Institutional Care and Use Committee (POSTECH IACUC) (Approval no. 2019-0088). All the mice are maintained in the animal facility of Pohang University of Science and Technology (POSTECH) Biotech Center in accordance with the Institutional Animal Care and Use Committee of POSTECH. All the animal experiments were performed according to the recommended guidelines.

### Cell culture and macrophage polarization for screening

RAW264.7 cell line was obtained from the Korean Cell Line Bank (KCLB, Seoul, Korea) (KCLB No. 40071), and used for screening. RAW 264.7 cells were cultured in 10 cm cell culture dish in Dulbecco's Modified Eagle Medium (DMEM with 4.5 g/L glucose, WELGENE), 10% Heat-Inactivated Fetal Bovine Serum (Gibco) and 1% Penicillin Strep-tomycin (WELGENE). The resting state of RAW264.7 cells were polarized into M1 or M2 cells by the addition of LPS (100 ng/mL, Sigma-Aldrich) and IFN-γ (20 ng/mL, R&D systems) for 24 h or M-CSF (20 ng/mL, R&D Systems), IL-4 (40 ng/mL, R&D Systems), and IL-13 (40 ng/mL, R&D Systems) for 6 days, respectively.

### Macrophage polarization from other cells

THP-1 cells were purchased by the Korean Cell Line Bank (KCLB, Seoul, Korea) (KCLB No. 40202) and they were grown in T75 flask in RPMI 1640 Medium (with 2.5 g/mL glucose, Gibco), 10% Heat-Inactivated Fetal Bovine Serum (Gibco), 1% Penicillin Streptomycin (WELGENE),

0.05% Sodium Pyruvate (Gibco) and 25 µM 2-Mercaptoethanol (Gibco). THP-1 cells were differentiated into macrophages by 100 nM of phorbol-12-myristate-13-acetate (Sigma-Aldrich), and differentiated into M1 by adding LPS (100 ng/mL, Sigma-Aldrich) and IFN-γ (20 ng/mL, R&D Systems) for 24 h, and M2 cells with M-CSF (20 ng/mL, Peprotech), IL-4 (40 ng/mL, R&D Systems), and IL-13 (40 ng/mL, R&D Systems) for 4 days. Primary peritoneal macrophages were collected from peritoneal cavity. The harvested cells were differentiated into M1 macrophages by LPS (100 ng/mL, Sigma-Aldrich) and IFN-γ (20 ng/mL, R&D Systems) and M2 cells with the addition of IL-4 (40 ng/mL, R&D Systems), and IL-13 (40 ng/mL, R&D Systems) for 24 h. The M0, M1, and M2 macrophages from THP-1 and mouse primary cells were stained with 0.5 µM of CDr17 for 30 min.

## High throughput screening
For the cell screening, M0, M1, and M2 from RAW264.7 cells were seeded into 96 well plates, and incubated with 1 µM of LC library in duplicate at 37 °C. After 1 h, cells were washed with PBS one time, and Operetta High Throughput Screening (Perkin Elmer, Walthan, MA, USA) was used to acquire the images from the screening. The data were collected from Harmony 4.8 (Perkin Elmer) and the fluorescent intensity was measured by the ImageJ (1.53) software, placing 10 regions of interest in randomly selected areas.

## Immunofluorescence staining
For the characterization of differentiated macrophages, APC anti-mouse CD86 Antibody (dilution 1:100, BioLegend, 105012), anti-mouse CD206 antibody (1:100, BIO-RAD, MCA2235) conjugated with Goat anti-Rat IgG (H + L) Cross-Adsorbed Secondary Antibody, Alexa Fluor 488 (1:500, Invitrogen, A-11006), Human CD38 Alexa Fluor® 488-conjugated Antibody (1:100, R&D Systems, FAB2404G), and CD36 Monoclonal Antibody (1:100, Invitrogen, MA5-14112) conjugated with Goat anti-Mouse IgG (H + L) Highly Cross-Adsorbed Secondary Antibody, Alexa Fluor Plus 488 (1:500, Invitrogen, A32723) were incubated with the cells for 30 min, and Hoechst33342 (1 µg/mL) was added for 5 min prior to the experiment. The images were taken by Operetta High Throughput Screening (Perkin Elmer, Walthan, MA, USA), and the data were collected from Harmony 4.8 (Perkin Elmer).

## D/L-glucose competition experiment
To measure the competitive effects of CDr17 with D/L-glucose, the media of M1 macrophages differentiated from RAW264.7 cell was changed into DMEM, no glucose (Gibco) with 10% Heat-Inactivated Fetal Bovine Serum (Gibco) and 1% Penicillin Streptomycin (WEL-GENE), containing 0, 20, 60 mM of D/L-glucose. After 5 min, CDr17 (1 µM) was added and incubated. The nuclues was stained with Hoechst33342 (1 µg/mL). After 30 min, the media was changed into fresh media. Fluorescence microscopy imaging was performed by Operetta High Throughput screening with ×40 objective lens. The data were collected from Harmony 4.8 (Perkin Elmer) and the fluorescent intensity was measured by the ImageJ (1.53) software, placing 10 regions of interest in randomly selected areas. The separate three experiments were proceeded with similar results. No data were excluded from the analyses.

## Induction of paw inflammation
Six to eight weeks old male wild-type C57BL/6 J were injected LPS (5 µg, 5 µL) and PBS (5 µL) into right and left paw respectively. After 24 h, redness and swelling were observed. 500 µM of CDr17 was intravenously injected, and the tissue was collected after 30 min.

## Induction of CAIA and in vivo imaging
Nine or ten weeks old female wild-type BALB/c were injected with a 2 mg Arthrogen-CIA-5-Clone monoclonal antibody Cocktail

(Chondrex) through intravenous (i.v.). After 3 days, 50 µg of lipopolysacchraide was injected intraperitoneally. On day 10, CDr17 (1 mM) was injected into CAIA-induced mice by i.v. injection. The in vivo imaging was performed by an AMI HTX (Spectral Instruments Imaging, Tucson, AZ, USA) with $\lambda_{ex} = 640$ nm and $\lambda_{em} = 690$ nm. The data were analyzed by Aura software (Spectral Instruments Imaging). The separate three experiments were proceeded with similar results. No data were excluded from the analyses.

## Induction of CIA and in vivo imaging
Nine weeks old female wild-type DBA1/j mice were purchased from Central Lab Animal Inc. (Seoul, Korea). 100 µL of emulsified solution (1:1, v/v) of collagen type II (4 mg/mL) (Chondrex) and CFA (1 mg/mL) (Chondrex) mixture was intradermally (i.d.) injected into DBA/1j mice. The incidence of arthritis was complete after 5 weeks, and CDr17 (1 mM) was injected into CIA mice by i.v. injection. The in vivo imaging was performed by an AMI HTX (Spectral Instruments Imaging, Tucson, AZ, USA) with $\lambda_{ex} = 640$ nm and $\lambda_{em} = 690$ nm. The data were analyzed by Aura software (Spectral Instruments Imaging). The separate three experiments were proceeded with similar results. No data were excluded from the analyses.

## Tissue imaging
The tissue samples from the animal model were harvested and frozen immediately. The frozen tissues were sectioned by the cryostat (Leica CM1850) with 10 µm thickness and sections were attached to the poly-L-lysine-coated slides. After the slides were washed with PBS, the samples were treated with FITC anti-mouse CD86 antibody (1:200, Biolegend, 105006), CD68 antibody (1:200, Biolegend, 130712), CD19 (1:50, eBioscience, 53-0194-82), CD11c (1:50, 43-044 for 30 min, GLUT1 (1:200, Invitrogen, MA5-31960) conjugated with Goat anti-Rabbit IgG (H + L) Highly Cross-Adsorbed Secondary Antibody, Alexa Fluor Plus 555 (1:500, Invitrogen, A32732) and Hoechst33342 (1 µg/mL) for 5 min prior to the experiment. The fluorescence images were obtained by Axio Observer (ZEISS, Oberkochen, Germany). The data were analyzed by the ImageJ (1.53) software.

## CRISPR-knockout experiment
The knockout experiment was carried out in M1 macrophages from RAW264.7 cell. TrueCut™ Cas9 Protein v2 (Invitrogen™, A36498) and Lipofectamine™ CRISPRMAX™ Cas9 Transfection Reagent (Invitrogen™, CMAX00008) were utilized according to the manufacture's guideline. The sgRNA in this experiment was purchased from Horizon (Target ID, SG-044254-01-0005; Target sequence, 5′-GGATGGGCTCTCCGTAGCGG-3′). The images were taken by Operetta High Throughput Screening (Perkin Elmer, Walthan, MA, USA), and the data were collected from Harmony 4.8 (Perkin Elmer). The fluorescent intensity was measured by the ImageJ (1.53) software, placing 10 regions of interest in randomly selected areas. The separate three experiments were proceeded with similar results. No data were excluded from the analyses.

## Isolation of B cells, dendritic cells, and T cells
The single cells were collected from the spleen, and they were lysed by RBC lysis buffer (Thermo Fisher Scientific, Rockford IL, USA). The collected cells were then incubated with biotinylated monoclonal antibodies for 15 min at 4°C. To isolate B cells, dendritic cells, and T cells, respectively, Mouse B Lymphocyte Set-DM (BD Bioscience Co., Franklin Lakes, NJ, USA), Mouse Dendritic Cell Enrichment Set-DM (BD Bioscience Co., Franklin Lakes, NJ, USA), and Mouse T Cell Enrichment Set-DM (BD Bioscience Co., Franklin Lakes, NJ, USA) were used respectively. The cells were washed with 1X BD IMag™ Buffer (10X buffer is diluted with deionized water, BD Bioscience Co., Franklin Lakes, NJ, USA) and then were centrifuged (1500 rpm, 5 min). Then, BD IMag™ Streptavidin Particle Plus – DM (BD Bioscience Co., Franklin

Lakes, NJ, USA) were added to cells bearing biotinylated antibodies. After 30 min, the tube containing the labeled cell suspension was placed within the magnetic field of the BD IMagnet™ (BD Bioscience Co., Franklin Lakes, NJ, USA) with IMag buffer.

## RT-PCR

The total RNA from M0, M1, and M2 macrophages was extracted using RNeasy Mini Kit (QIAGEN Inc., 74106), and the amount and quality were measured by Nanodrop 2000 (Thermo Scientific). cDNA was synthesized with a High Capacity cDNA Reverse Transcription Kit (Applied Biosystems) according to the instruction. The qRT-PCR was carried out by TB Green™ Premix Ex Taq™ II (Tli RNaseH Plus) Kit (TaKaRa). The reactions were run on a qTOWER[3] Real-time PCR system (Analytic Jena) with the following cycles: 10 min at 95 °C, and 40 cycles of 15 s at 95 °C and 1 min at 55 °C. The experiment was repeated three times individually. No data were excluded from the analyses. The data were collected by qPCRsoft 4.0 (analytic jena). The designed primers were displayed in Supplementary Table 3.

## Establishment of HeLa-GLUT1-CRISPRa and CRISPRi system

For the experiment, HeLa cells were purchased from the Korean Cell Line Bank (KCLB, Seoul, Korea) (KCLB No. 10002). The slc2a1 Lentiviral sgRNA (Dharmacon, VSGH11888-247233466, 5′-AGAGCGCCGCCCAG GAC-3′) was infected to $5 \times 10^4$ of HeLa cells expressed with VPR and KRAB, and the cells were seeded into a 24-well plate one day before infection. After 2 days, the selection was conducted using 3 µg/mL of puromycin. On day 7, the cells were stained with anti-GLUT1 (R&D Systems, MAB1418, 1:500) conjugated with Goat anti-Mouse IgG (H + L) Highly Cross-Adsorbed Secondary Antibody, Alexa Fluor Plus 488 (1:500, Invitrogen, A32723) and checked the intensity by flow cytometry (LSR Foretessa 5 laser Flow Cytometer, BD, NJ, USA). 5% of brightest or dimmest from VPR or KRAB, respectively, were sorted and seeded into 96-well plates for single-cell cloning. The cells were collected and the GLUT1 expression was confirmed by flow cytometry with anti-GLUT1Cells were analyzed on an LSR Fortessa II (BD Biosciences), and data were processed with FlowJo software. The western blot was proceeded to double check the expression level. The cells were washed with PBS, and lysed by M-PER™ Mammalian Protein Extraction Reagent (Thermo Scientific) with Halt™ Protease Inhibitor Cocktail (Thermo Scientific). Bradford assay was used to measure the protein concentration. Anti-GLUT-1 antibody (1:1000, Sigma, SAB5500114) and β-actin (1:1000, Santa Cruz, sc-47778) conjugated with Goat anti-Mouse IgG (H + L) Highly Cross-Adsorbed Secondary Antibody, Alexa Fluor Plus 488 (1:500, Invitrogen, A32723) were used for labeling. Membranes were washed three times with TPBS and incubated 1 h with secondary antibody-Alex647 (1:1000). The protein bands were detected and quantified by Chemidoc MP Imaging System (Bio-Rad, CA, USA). After establishment, the cells were incubated with CDr17 for 30 min, and then imaged. The images were taken by Operetta High Throughput Screening (Perkin Elmer, Walthan, MA, USA), and the data were collected from Harmony 4.8 (Perkin Elmer). The separate three experiments were proceeded with similar results. No data were excluded from the analyses.

## Statistics and reproducibility

For our research, we repeated two or three times of the experiments to secure the reproducibility. During the experiment, the samples were randomized into different groups, and investigators were blinded to group allocation. Statistical analysis was performed with Graphpad Prism software (La Jolla, USA), and no data were excluded from the analyses. All the data are presented as mean ± SD.

## Reporting summary

Further information on research design is available in the Nature Research Reporting Summary linked to this article.

## Data availability

The authors declare that all data that support our findings in this study are included in the manuscript and supplementary information. Source data are provided with this paper.

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

## Acknowledgements

This research was supported by the Institute for Basic Science (IBS) (IBS-R007-A1 to Y.-T.C. and H.-Y.K.), the National Research Foundation of Korea (NRF) grant funded by the Korea government (MSIT) (2020R1A2C2009776 to N.-Y.K.), the Ministry of Education (2020R1A6A1A03047902 to N.-Y.K.), the intramural fund of POSTECH (Pohang University of Science and Technology, to N.-Y.K.), and Hyundai Motor Chung Mong-Goo Foundation scholarship to H.C.

## Author contributions

H.C. synthesized library, and S.H.L., X.L. characterized CDr17 probe. H.C. did screening for M1 selective probe discovery, and performed cell culture experiments. H.C., H.-Y.K., A.S., and S.-H.I. did animal experiments. J.-J.K. and N.M. prepared CRISPRa- and CRISPRi-GLUT1 platform for CDr17 mechanism study. H.C. and H.-Y.K. discovered CDr17 mechanism. H.C., H.-Y.K., N.-Y.K. and Y.-T.C. reviewed, analyzed, and interpreted the data. H.C., H.-Y.K. and Y.-T.C. wrote the paper. All authors discussed the results and commented on the manuscript.

## Competing interests

Y.-T.C., H.-Y.K., and H.C. have submitted a patent application to the Korean patent office pertaining to "Visualizing inflammation with M1 macrophage selective probe via GLUT1 as the gating target" of this work (application number 10-2022-0039430). S.-H.I. is the CEO of Immuno-Biome but declares no competing interests for this paper. The remaining authors declare no competing interests.
