## [Peer Review File · Nature Communications]

REVIEWER COMMENTS

Reviewer #1 (Remarks to the Author):

I take the point about the importance of macrophage imaging by finding a molecular probe in inflammation. Yet, it is my view that there remain some distance between the contents of this study and substantial impact on the novelty and creativity for that particular perspective to carry this paper for Nature Communications. Some points are indicated below:

1) Over hundreds of Cyanine-based glycoconjugate fluorescent probes targeting GLUT1 transporters were designed, discovered and published. In addition, selective expression of GLUT1 in M1 phenotype of macrophage has been well studied as a known phenomenon (THE JOURNAL OF BIOLOGICAL CHEMISTRY VOL. 289, NO. 11, pp. 7884 –7896). What outstanding characteristics make CDr17 a perfect GLUT1 imaging probe especially in the application for inflammation detection?

2) In the current and their previous publications, the author advocate that the selective staining mechanism of the GLUT-targeted bioprobe (CDr17 in this study) is act as “gating target” or Gating-Oriented Live-cell Distinction (GOLD) mechanism, which is different from either the Protein Oriented-Live cell Distinction (POLD) or holding-oriented live-cell distinction (HOLD) mechanisms (Acc. Chem. Res. 2019, 52, 11, 3097–3107, Chem. Soc. Rev., 2022,51, 1573-1591) However, other researches have reported that the GLUT1 transmembrane protein can undergo intracellular internalization and trafficking between internal vesicular compartments and the cell surface (ACS Omega 2020, 5, 26, 15911–15921). More studies have revealed that GLUT1 activity, recycling, and internalization are regulated aspects of growth factor-stimulated glucose uptake (Mol Biol Cell. 2007;18(4):1437-1446). These results are supportive for the concept that all GLUT1-targeting probes can be explained just as a “GLUT1 binding molecule” which may travel together with the internalized transporters rather than just been transported into the cytoplasm through GLUT1.

3) The current study can not exclude the possibility of CDr17 (and other dye molecules) is an inhibitor of GLUT1. Some studies shown that GLUT1 inhibitor is more effective for GLUT staining compared to its substrate, as the authors also found that CDr17 is much sensitive compared to 2-NBDG in this study.

4) The authors have also published non-glycoconjugated cyanine probes that are “powerful fluorescence imaging probe for macrophage-targeted inflammation imaging” (PLoS ONE 2014,9(7): e103721; Chem. Commun., 2014, 50, 6589--6591). How to believe which one is the best? Any cyanine dye? or only the glycoconjugated cyanine dye? This is also relative to the query in 1).

Reviewer #2 (Remarks to the Author):

This is a critical review of “Visualizing inflammation with M1 macrophage selective 1 probe via GLUT1 as the gating target” by Heewon Cho et al. This manuscript describes the randomized selection of a carbohydrate-NIR dye conjugate as selective for the GLUT1 transporter. The team has done very nice work to characterize the subtype selectivity of this conjugate although a K_i was not disclosed for either mouse or human transporters. Furthermore, the researchers have described increased GLUT1 expression (and activity?) in M1 polarized versus M0 or M2 in in vitro settings. Blocking studies with appropriate compounds were done and satisfy this reviewer’s belief in molecular specificity. Similarly, uptake in living, polarized cells was done along with uptake in living mice activated using a Chondrex kit. Joint fluorescence was proportional to joint swelling. New probes for targeted immune cell types are badly needed and probes targeting specific polarization status of macrophages has been lacking. The work here is promising but lacks the evidence needed to assign probe uptake to M1 macrophages in vivo. This reviewer cites the following deficiencies:

1. Please have a native speaker of English edit the manuscript for grammatical errors.
2. Cy5 has an emission max of 660 nm. This will not be suitable for the clinic due to scattering and absorption by tissue.
3. The Chondrex kit uses a pulse of LPS days after primary antibody administration. This LPS pulse is why your team is observing M1 polarized macrophages. If your team had used the long collagen II antigen method, by the time RA developed, you would have native polarizations not influenced by LPS.
4. CD86 is a marker for macrophages, dendritic cells and B cells, all of which are densely localized to the RA joint. They likely all express GLUT1. In order to prove selectivity for M1 macrophages, co-staining for B and dendritic cells should also be shown. This is critical from the fluorescent joint to make this claim of specificity for M1 macrophages.
5. Line 78 typo “Cy52”
6. Line 147 “It is known...” Please cite a reference. Reference 33 does not support this statement.
7. Line 153 ...”showed highest peak at 15 min” At 15 minutes, the conjugate is still widely circulating in the blood pool and inflamed tissues get more blood volume than non-inflamed tissues. It would be better to have conducted in vivo blocking studies. If 15 minutes post-injection is real binding, it should be blocked by a specific inhibitor or reduced by a bolus of glucose. FDG has the benefit of being trapped once imported. Is there evidence of trapping of CDr17?
8. Figure 4F. This reviewer knows the difficulty of cutting frozen knees without fixation or decalcification. The CD86 staining is very poor and the CDr17 retention does not appear to be intracellular. Fixed, decalcified slides should be made and co-stained for pan-macrophage, DCs and Bs to work out which cell types might also be retaining CDr17 through GLUT1 activity.

Reviewer #3 (Remarks to the Author):

In this manuscript, Cho and colleagues develop and characterize a macrophage-selective fluorescent probe (CDr17) that is selective for the M1 subtype in mouse and human macrophage cell lines, as well as primary mouse macrophages. The selectivity of CDr17 is ascribed to the relative overexpression of the glucose transport GLUT1 in the M1 subtype. CDr17 fluorescence is also demonstrated in a rheumatoid arthritis model. This probe complements the groups previous efforts to develop a similar dye, which was instead selective for activated macrophages, irrespective of subtype.

The authors are addressing an important technical barrier, with the result that the work has applicability to a broad readership. CDr17 represents a conceptual advance over qPCR/antibody labeling-based means of characterizing macrophage phenotype. The authors show via multiple means that GLUT-1 function is necessary, sufficient and selective as an M1-specific target. The manuscript is well-written and thorough in its execution. The D/L-glucose experiments to demonstrate GLUT-1 dependence are particularly elegant. That said, I have some questions:

1. M1/M2 is a useful distinction with in vitro polarization, but macrophages in vivo are more likely to occupy a position along a continuous spectrum between these two extremes. With this in mind, could the authors comment on whether they would expect a correlation between the strength of M1 polarization and CDr17 intensity? Or, phrased another way, does the expression level of GLUT-1 correlate with the strength of M1 polarization?

2. Could the authors comment on whether CDr17 could be delivered repeatedly to the same animal over the course of RA inflammation, such that a decrease in CDr17 signal (i.e. a shift away from M1-like populations at the site of inflammation) could be captured over time?

3. It would be helpful to know the degree to which other, non-macrophage cell types express GLUT-1 in vivo, in order to interpret the RA data accurately. CDr17 signal was detected in CD86+ cells, but they may not be the sole cell type exhibiting CDr17 uptake in inflamed tissues.

I also have some minor typographical issues, which I have outlined below:

Line 59: 'In [recent] decades' was perhaps the intended meaning.

Line 125: consider substituting 'prepared for' with 'generated'.

Line 169: 'position' is misspelled.

Line 198: I assume the authors used 10 cm cell culture dishes, rather than 10 mm?

Line 206: cells were 'grown'

Line 246: The use of the word 'enucleated' is somewhat confusing – would 'harvested' be a more suitable replacement?

Line 247: 'Sections were attached to the poly-L-lysine-coated slides' is a clearer description.

Line 302: would it be more correct to say 'CDr17 is the subject of a pending patent application'?

Reviewer #1 (Remarks to the Author):

I take the point about the importance of macrophage imaging by finding a molecular probe in inflammation. Yet, it is my view that there remain some distance between the contents of this study and substantial impact on the novelty and creativity for that particular perspective to carry this paper for Nature Communications. Some points are indicated below:

Answer: We thank the reviewer for valuable comments and suggestions. The detailed response is listed below.

1) Over hundreds of Cyanine-based glycoconjugate fluorescent probes targeting GLUT1 transporters were designed, discovered and published. In addition, selective expression of GLUT1 in M1 phenotype of macrophage has been well studied as a known phenomenon (THE JOURNAL OF BIOLOGICAL CHEMISTRY VOL. 289, NO. 11, pp. 7884 –7896). What outstanding characteristics make CDr17 a perfect GLUT1 imaging probe especially in the application for inflammation detection?

Answer: We agree with the reviewer's point that many developed cyanine-based glycoconjugate fluorescent probes have been developed. Their application, however, mostly focused on imaging tumors with known GLUT1 overexpression, and few of them were used to immune cell distinction, especially macrophages.

As commented by the reviewer, though GLUT1 overexpression in M1 macrophages has been well investigated, studies to develop probes targeting M1 macrophages have been lacking. To develop an outstanding M1-specific probe, we systemically established the LC library. Following our experiments, we found an M1-specific probe, **CDr17**, using GLUT1 as the main gate to enter M1 macrophages. To the best of our knowledge, **CDr17** firstly proved its potential to detect M1 macrophages in inflammation. We believed that the above view brought novelty to **CDr17**.

2) In the current and their previous publications, the author advocate that the selective staining mechanism of the GLUT-targeted bioprobe (CDr17 in this study) is act as “gating target” or Gating-Oriented Live-cell Distinction (GOLD) mechanism, which is different from either the Protein Oriented-Live cell Distinction (POLD) or holding-oriented live-cell distinction (HOLD) mechanisms (Acc. Chem. Res. 2019, 52, 11, 3097–3107, Chem. Soc. Rev., 2022,51, 1573-1591) However, other researches have reported that the GLUT1 transmembrane protein can undergo intracellular internalization and trafficking between internal vesicular compartments and the cell surface (ACS Omega 2020, 5, 26, 15911–15921). More studies have revealed that GLUT1 activity, recycling, and internalization are regulated aspects of growth factor-stimulated glucose uptake (Mol Biol Cell. 2007;18(4):1437-1446). These results are supportive for the concept that all GLUT1-targeting probes can be explained just as a “GLUT1 binding molecule” which may travel together with the internalized transporters rather than just been transported into the cytoplasm through GLUT1.

Answer: As suggested by the reviewer, we demonstrated whether **CDr17** undergoes intracellular internalization by binding to GLUT1 through the checking of **CDr17** localization in the cell. It turned out that **CDr17** was located in mitochondria, which is not relevant to the GLUT1 intracellular internalization process. Furthermore, the direct localization correlation of **CDr17** with GLUT1 is quite low (0.68). This data indicates that **CDr17** is transported into the

cytoplasm through GLUT1 on the membrane, not co-internalized by binding to GLUT1. Therefore, the mechanism can be defined by Gating-Oriented Live-cell Distinction (GOLD).

A) M1 macrophages were stained with Mitotracker Orange (MTO, 100 nM, Green) for 15 min, and **CDr17** (1 μ M) were treated in the cells for 20 min. Live state cells were imaged by fluorescence microscopy after washing the cells. Then, the cells were fixed and permeabilized before incubation with the GLUT1 antibody. B) The correlation between **CDr17** and mitotracker or GLUT1 was analyzed and displayed. The numeric values represented the Pearson correlation coefficient. C) The correlation of ROI (white arrow in A) was interpreted. Scale bar; 20 μ m. Pseudo color: **CDr17** (red), Mitotracker (green), and GLUT1 (cyan).

3) *The current study can not exclude the possibility of CDr17 (and other dye molecules) is an inhibitor of GLUT1. Some studies shown that GLUT1 inhibitor is more effective for GLUT staining compared to its substrate, as the authors also found that CDr17 is much sensitive compared to 2-NBDG in this study.*

Answer: As pointed out by the reviewer, we have clarified whether **CDr17** acts as an inhibitor, by staining M1 macrophages with **CDr17** and 2-NBDG together. The results showed that **CDr17** did not meaningfully affect 2-NBDG staining patterns in M1 macrophages. With this result, we concluded that **CDr17** utilizes GLUT1 for the entry to get inside M1 macrophages, not acting as a GLUT1 inhibitor.

A) 2NBDG (100 μ M) and B) **CDr17** (1 μ M) alone stained M0 and M1 macrophages for 30 min. C) **CDr17** (1 μ M) and 2NBDG (100 μ M) together stained M0 and M1 macrophages for 30 min. Scale bars, 20 μ m.

4)The authors have also published non-glycoconjugated cyanine probes that are “powerful fluorescence imaging probe for macrophage-targeted inflammation imaging” (*PLoS ONE* 2014,9(7): e103721; *Chem. Commun.*, 2014, 50, 6589--6591). How to believe which one is the best? Any cyanine dye? or only the glycoconjugated cyanine dye? This is also relative to the query in 1).

Answer: Our group has published papers on macrophage-targeting probes, and made many efforts to develop fluorescent probes surpassing previous ones. Though the published probes can visualize macrophages in inflammation *in vivo*, they could not distinguish between M1 and M2 macrophages. The molecular targets were also not identified. Therefore, **CDr17** is superior to previously published probes, given targeting M1 macrophages with clear mechanism elucidation.

In addition, considering the features of M1 macrophages, we established a luminescent-carbohydrate library, which would be helpful to give a systematic comparison and higher specificity to M1 macrophages. Based on the results, carbohydrate-conjugated fluorophores will be better than fluorophores alone in developing M1-specific probes.

Reviewer #2 (Remarks to the Author):

This is a critical review of “Visualizing inflammation with M1 macrophage selective 1 probe via GLUT1 as the gating target” by Heewon Cho et al. This manuscript describes the randomized selection of a carbohydrate-NIR dye conjugate as selective for the GLUT1 transporter. The team has done very nice work to characterize the subtype selectivity of this conjugate although a K_i was not disclosed for either mouse or human transporters. Furthermore, the researchers have described increased GLUT1 expression (and activity?) in M1 polarized versus M0 or M2 in in vitro settings. Blocking studies with appropriate compounds were done and satisfy this reviewer’s belief in molecular specificity. Similarly, uptake in living, polarized cells was done along with uptake in living mice activated using a Chondrex kit. Joint fluorescence was proportional to joint swelling. New probes for targeted immune cell types are badly needed and probes targeting specific polarization status of macrophages has been lacking. The work here is promising but lacks the evidence needed to assign probe uptake to M1 macrophages in vivo. This reviewer cites the following deficiencies:

Answer: We thank the reviewer for the positive and constructive comments. A detailed response is listed below.

1. Please have a native speaker of English edit the manuscript for grammatical errors.

Answer: We carefully checked the errors and corrected them throughout the manuscript.

2. Cy5 has an emission max of 660 nm. This will not be suitable for the clinic due to scattering and absorption by tissue.

Answer: We agree that the emission wavelength of Cy5 is relatively shorter than the clinical NIR probe. It would be better to have a longer wavelength in the probe, Cy7 dyes did not show high enough selectivity to M1. Still, there are references that Cy5 could be utilized in clinical samples due to relatively low tissue scattering and superior brightness to other NIR dyes (PMID: 32637234; PMID: 28721307). In our manuscript, **CDr17** also clearly visualized macrophages at the tissue level. For these reasons, **CDr17** may yet have the potential to be used in clinical samples.

3. The Chondrex kit uses a pulse of LPS days after primary antibody administration. This LPS pulse is why your team is observing M1 polarized macrophages. If your team had used the long collagen II antigen method, by the time RA developed, you would have native polarizations not influenced by LPS.

Answer: As suggested by the reviewer, we tested the **CDr17** selectivity in the (LPS-free) collagen-induced arthritis (CIA) models. We confirmed that **CDr17** also successfully visualized the inflamed part in the CIA model, a similar result to the LPS-involved model. In summary, **CDr17** can detect inflammation in arthritis regardless of LPS usage.

Supplementary Fig. 16. CDr17 selectivity in CIA animals. A) Workflow of collagen-induced arthritis (CIA) model preparation and in vivo imaging. (CII: Collagen Type II; CFA: complete Freund's adjuvant) B, C) The CDr17 intensity from RA (n=6) by cont (n=4) animals was displayed. The red arrow represented the swelling joint part.

We updated the results in the revised manuscript (page 4, lines 153-156) and added the detailed data in the supplementary information.

4. *CD86 is a marker for macrophages, dendritic cells, and B cells, all of which are densely localized to the RA joint. They likely all express GLUT1. In order to prove selectivity for M1 macrophages, co-staining for B and dendritic cells should also be shown. This is critical from the fluorescent joint to make this claim of specificity for M1 macrophages.*

Answer: We thank the reviewer for the excellent suggestion, also mentioned by Reviewer 3. As commented by the reviewer, more suitable antibody sets were used for the RA tissue imaging: macrophages (CD68), B cells (CD19), dendritic cells (CD11c) and GLUT1 antibody. The improved images clearly showed that CDr17-staining cells were mostly macrophages, with higher expression levels of GLUT1 than other cell types. To clarify the point, we isolated B cells, dendritic cells, and T cells from the mouse, analyzed their GLUT1 expression levels, and compared them with M0 and M1 macrophages. The result indeed demonstrated that M1 macrophages have higher expression of GLUT1 than any other cell type.

Supplementary Fig. 18. CDr17 specificity to M1 macrophages compared to other cell types in synovium tissues from RA animals. Antibody: a) CD68, b) CD11c, c) CD19. CDr17 strongly stained CD68-positive cells than CD11c and CD19-positive cells, and they have higher expression levels of GLUT1. Scale bar: 20 μ m

Supplementary Fig. 19. M1 macrophages have higher GLUT1 expression. Each cell type was confirmed with their specific biomarkers (CD19, CD11c, CD3) respectively, and compared GLUT1 level among them.

We have updated the results in the revised manuscript (page 4, lines 164-171) and added the detailed data in the supplementary information.

5. Line 78 typo “Cy52”

Answer: As suggested by the reviewer, we have corrected the typo.

6. Line 147 “It is known...” Please cite a reference. Reference 33 does not support this statement.

Answer: As pointed out by the reviewer, we have included and changed the reference into the correct one. (PMID: 32530555)

7. Line 153 ...”showed highest peak at 15 min” At 15 minutes, the conjugate is still widely circulating in the blood pool and inflamed tissues get more blood volume than non-inflamed tissues. It would be better to have conducted in vivo blocking studies. If 15 minutes post-

injection is real binding, it should be blocked by a specific inhibitor or reduced by a bolus of glucose. FDG has the benefit of being trapped once imported. Is there evidence of trapping of CDr17?

Answer: We thank the reviewer for this comment. As suggested by the reviewer, we did *in vivo* glucose competition assay with RA animals. CDr17 was injected (i.v.) into the mouse, followed by the glucose i.v. injection. At 15 min, we observed that the CDr17 signal decreased in the glucose-treated mouse compared to non-treated ones (around a 2-fold difference).

Glucose competition assay *in vivo*. Glucose (2 M, 150 μ L) was intravenously (i.v.) injected into the mouse before 5 min to i.v. injection of CDr17 (250 μ M, 100 μ L). The left was not administered by the glucose, but the right was. The images were taken after 15 min post-injection of CDr17.

8. Figure 4F. This reviewer knows the difficulty of cutting frozen knees without fixation or decalcification. The CD86 staining is very poor and the CDr17 retention does not appear to be intracellular. Fixed, decalcified slides should be made and co-stained for pan-macrophage, DCs and Bs to work out which cell types might also be retaining CDr17 through GLUT1 activity.

Answer: We thank the reviewer for this comment. As suggested by the reviewer, fixation, and decalcification were tried and costained with antibodies. Though we observed the antibody signals clearly, it was hard to recognize the CDr17 signal under this condition. Instead, from frozen joint parts (without fixation and decalcification) (PMID: 24482171), we could get the CDr17 signal back as well as antibodies. In conclusion, the fixation or decalcification step does not help the performance of CDr17, but actually weakens the signal.

The synovium tissue images a) with fixation and decalcification and b) without fixation and decalcification under the same contrast. Scale bar: 20 μ m.

Reviewer #3 (Remarks to the Author):

In this manuscript, Cho and colleagues develop and characterize a macrophage-selective fluorescent probe (CDr17) that is selective for the M1 subtype in mouse and human macrophage cell lines, as well as primary mouse macrophages. The selectivity of CDr17 is ascribed to the relative overexpression of the glucose transport GLUT1 in the M1 subtype. CDr17 fluorescence is also demonstrated in a rheumatoid arthritis model. This probe complements the groups previous efforts to develop a similar dye, which was instead selective for activated macrophages, irrespective of subtype.

The authors are addressing an important technical barrier, with the result that the work has applicability to a broad readership. CDr17 represents a conceptual advance over qPCR/antibody labeling-based means of characterizing macrophage phenotype. The authors show via multiple means that GLUT-1 function is necessary, sufficient and selective as an M1-specific target. The manuscript is well-written and thorough in its execution. The D/L-glucose experiments to demonstrate GLUT-1 dependence are particularly elegant. That said, I have some questions:

Answer: We thank the reviewer for the positive and constructive comments. A detailed response is listed below.

1. M1/M2 is a useful distinction with in vitro polarization, but macrophages in vivo are more likely to occupy a position along a continuous spectrum between these two extremes. With this in mind, could the authors comment on whether they would expect a correlation between the strength of M1 polarization and CDr17 intensity? Or, phrased another way, does the expression level of GLUT-1 correlate with the strength of M1 polarization?

Answer: We thank the reviewer for this suggestion. The intensity from **CDr17**, GLUT1, and CD86 displayed a linear relationship following the progress of polarization. This result indicated a strong correlation between M1 polarization and **CDr17** or GLUT1.

Supplementary Fig. 12. Correlation between M1 polarization, **CDr17**, and GLUT1. RAW264.7 cells were polarized into M1 macrophages with the treatment of LPS (100 ng/mL) and IFN- γ (20 ng/mL). During polarization, the intensity of **CDr17**, GLUT1, and CD86 was checked at 3 h intervals until 24 h. The cells were firstly stained with **CDr17** (1 μ M, 30 min), and were fixed and permeabilized. The fixed cells were incubated with antibodies. Scale bar: 20 μ m.

We have updated the results in the revised manuscript (page 3, lines 113-116) and added the detailed data in the supplementary information.

2. *Could the authors comment on whether CDr17 could be delivered repeatedly to the same animal over the course of RA inflammation, such that a decrease in CDr17 signal (i.e. a shift away from M1-like populations at the site of inflammation) could be captured over time?*

Answer: Following the suggestion, we repeatedly injected **CDr17** into the same animals. The **CDr17** intensity decreased as time passed, and the signal from the disease-induced mouse was similar to that from the controls on day 28. Also, we observed that severe redness and swelling of the joint reached maximally on 12 days, and the symptoms continuously declined. With these results, we concluded that **CDr17** intensity decreased following the weakened severity grade, and it did not accumulate in the region despite injection over time.

Supplementary Fig. 17. Tracking the CDr17 signal in RA animals over time injection. A) Workflow of collagen antibody-induced arthritis (CAIA) model preparation and in vivo imaging. The star was marked to represent the day for the maximum severity, and the red triangle showed when in vivo images were taken. The CDr17 intensity from RA (n=3) by cont (n=2) animals was displayed in B) the graph, and C) images. D) The joint swelling grade was decreased after reaching the maximum score following the days.

We have updated the results in the revised manuscript (page 4, lines 158-163) and added the detailed data in the supplementary information.

3. It would be helpful to know the degree to which other, non-macrophage cell types express GLUT-1 in vivo, in order to interpret the RA data accurately. CDr17 signal was detected in CD86+ cells, but they may not be the sole cell type exhibiting CDr17 uptake in inflamed tissues.

Answer: We thank the reviewer for the excellent suggestion, also mentioned by Reviewer 2. As commented by the reviewer, more suitable antibody sets were used for the RA tissue imaging: macrophages (CD68), B cells (CD19), dendritic cells (CD11c), and GLUT1 antibody. The improved images clearly showed that CDr17-staining cells were mostly macrophages,

with higher expression levels of GLUT1 than other cell types. To clarify the point, we isolated B cells, dendritic cells, and T cells from the mouse, analyzed their GLUT1 expression levels, and compared them with M0 and M1 macrophages. The result indeed demonstrated that M1 macrophages have higher expression of GLUT1 than any other cell type.

Supplementary Fig. 18. CDr17 specificity to M1 macrophages compared to other cell types in synovium tissues from RA animals. Antibody: a) CD68, b) CD11c, c) CD19. CDr17 strongly stained CD68-positive cells than CD11c and CD19-positive cells, and they have higher expression levels of GLUT1. Scale bar: 20 μ m

Supplementary Fig. 19. M1 macrophages have higher GLUT1 expression. Each cell type was confirmed with their specific biomarkers (CD19, CD11c, CD3) respectively, and compared GLUT1 level among them.

We have updated the results in the revised manuscript (page 4, lines 164-171) and added the detailed data in the supplementary information.

I also have some minor typographical issues, which I have outlined below:

Line 59: 'In [recent] decades' was perhaps the intended meaning.

Answer: As suggested by the reviewer, we have replaced the expression with 'In recent decades'.

Line 125: consider substituting 'prepared for' with 'generated'.

Answer: As pointed out by the reviewer, we substituted 'prepared for' with 'generated'.
Line 169: 'position' is misspelled.

Answer: We have corrected the word, as pointed out by the reviewer.

Line 198: I assume the authors used 10 cm cell culture dishes, rather than 10 mm?

Answer: We have edited the unit correctly as commented by the reviewer.

Line 206: cells were 'grown'

Answer: We have changed the word as pointed out by the reviewer.

Line 246: The use of the word 'enucleated' is somewhat confusing – would 'harvested' be a more suitable replacement?

Answer: As pointed out by the reviewer, we have changed the word, 'enucleated' into 'harvested'.

Line 247: 'Sections were attached to the poly-L-lysine-coated slides' is a clearer description.

Answer: As suggested by the reviewer, we have changed the description.

Line 302: would it be more correct to say 'CDr17 is the subject of a pending patent application'?

Answer: We have clarified the sentence as suggested by the reviewer.

REVIEWERS' COMMENTS

Reviewer #1 (Remarks to the Author):

The novel contributions, and relation to the authors' previous work are explicitly stated. The authors positively addressing the concerns in my original review.

Reviewer #2 (Remarks to the Author):

The authors have made many improvements to the manuscript and have satisfied most of this reviewer's previously listed concerns. The following concerns are noted:

1. Supplementary figure 15 slides still look unfocused.
2. Supplementary fig 16. CDr17 is injected on day 6. Why does data collection start on day 12 in Supplementary Fig. 17?
3. Supplementary Fig 18. The staining is convincing.
4. The 15 min uptake time for CDr17 is still very short and indicative of blood pool in inflamed tissue. Labeled albumin would show the same/similar result. Why was in vivo blocking with STF-31 not done?
5. Cy5 has excitation/emission peaks of 640/670 nm. Even in pre-clinical models, attenuation of both wavelengths is significant. Because the dye is part of the pharmacophore, changes to more red-shifted dyes isn't possible, limiting this new probe to small, shaved animals.

Reviewer #3 (Remarks to the Author):

Many thanks to the authors for their comprehensive addressing of my questions and concerns.

The manuscript has been strengthened by the addition of data suggesting that CDr17 is transported into the cytoplasm, rather than co-internalized with Glut-1, and overlaps with CD68+ macrophages, as opposed to CD11c/GLUT1.

The addition of collagen-induced arthritis data and demonstrated competition with i.v. glucose, both of which add rigor to the study.

I have no further queries or revisions to suggest.

REVIEWERS' COMMENTS

Reviewer #1 (Remarks to the Author):

The novel contributions, and relation to the authors' previous work are explicitly stated. The authors positively addressing the concerns in my original review.

Answer: We thank the reviewer for constructive comments and positive feedback.

Reviewer #2 (Remarks to the Author):

The authors have made many improvements to the manuscript and have satisfied most of this reviewer's previously listed concerns. The following concerns are noted:

1. Supplementary figure 15 slides still look unfocused.

Answer: Following the reviewer's comment, we improved the images with more careful focusing.

Supplementary Fig. 15. Tissue section images of CDr17 from the acute inflammation-induced animal model after i.v. injection of CDr17. All the images were acquired at 20x magnification. Each three experiments was repeated independently with similar results. Scale bar, 20 μ m.

2. Supplementary fig 16. CDr17 is injected on day 6. Why does data collection start on day 12 in Supplementary Fig. 17?

Answer: Following the reviewer's comment, we made the collagen-induced arthritis models (CIA) in Supplementary Fig. 16, which takes about 6 weeks (not 6 days). Supplementary Fig. 17 is based on collagen antibody induced arthritis method (CAIA), which takes about 12 days. Due to the difference of the models, CDr17 injection was performed at 6 weeks and 12 days, respectively.

3. Supplementary Fig 18. The staining is convincing.

Answer: We appreciate the reviewer for the encouraging comment.

4. The 15 min uptake time for CDr17 is still very short and indicative of blood pool in inflamed tissue. Labeled albumin would show the same/similar result. Why was in vivo blocking with STF-31 not done?

Answer:

1) The one-pass circulation time of blood in a mouse was previously determined to be approximately 15 s (*J. Histochem. Cytochem.* **1998**, *46*, 627-639.), so we assumed 15 min is a reasonably long time to allow the tissue localization of the probe.

2) At 15 min, CDr17 accumulation in the target area was maximum both for inflamed and control animal, and the ratio was maintained almost same in the prolonged time period. As the decreased intensity caused bigger error range, we selected 15 min as the best time point for imaging.

3) We tried STF-31 in the experiment, but unfortunately, it was hard to set the optimum conditions with limited references for the in vivo experiments. At least, the alternative glucose competition approach was successful. Also, we showed a clear cellular inhibition results by STF-31, proving the concept of the mechanism.

5. Cy5 has excitation/emission peaks of 640/670 nm. Even in pre-clinical models, attenuation of both wavelengths is significant. Because the dye is part of the pharmacophore, changes to more red-shifted dyes isn't possible, limiting this new probe to small, shaved animals.

Answer: As the reviewer mentioned, we agree that extending the conjugation of CDr17 to longer wavelength dye may not be easy due to the pharmacophore role of Cy5. Despite its shorter wavelength than NIR dyes, as it shows a superior brightness, we believe Cy5 still could be utilized in clinical samples with proper optical setting. At least, in our experiment, we did not shave the animals, but still we could get the high quality of in vivo images.

Reviewer #3 (Remarks to the Author):

Many thanks to the authors for their comprehensive addressing of my questions and concerns.

The manuscript has been strengthened by the addition of data suggesting that CDr17 is transported into the cytoplasm, rather than co-internalized with Glut-1, and overlaps with CD68+ macrophages, as opposed to CD11c/GLUT1.

The addition of collagen-induced arthritis data and demonstrated competition with i.v. glucose, both of which add rigor to the study.

I have no further queries or revisions to suggest.

Answer: We thank the reviewer for constructive comments and positive feedback.